# Characterizing Mechanical, Heat Seal, and Gas Barrier Performance of Biodegradable Films to Determine Food Packaging Applications

**DOI:** 10.3390/polym14132569

**Published:** 2022-06-24

**Authors:** Bram Bamps, Rafael Moreno Macedo Guimaraes, Gwen Duijsters, Dries Hermans, Jan Vanminsel, Evelynn Vervoort, Mieke Buntinx, Roos Peeters

**Affiliations:** 1Materials and Packaging Research & Services, Imo-Imomec, Hasselt University, Wetenschapspark 27, 3590 Diepenbeek, Belgium; gwen.duijsters@student.uhasselt.be (G.D.); dries.hermans@student.uhasselt.be (D.H.); jan.vanminsel@student.uhasselt.be (J.V.); evelynn.vervoort@student.uhasselt.be (E.V.); mieke.buntinx@uhasselt.be (M.B.); roos.peeters@uhasselt.be (R.P.); 2Instituto de Química, Universidade de São Paulo, Avenida Professor Lineu Prestes, 748 Butanta, São Paulo 05508-070, Brazil; rafammguima@usp.br

**Keywords:** biodegradable food packaging, heat sealing, mechanical characterization, seal through contamination, coated paper

## Abstract

In an organic circular economy, biodegradable materials can be used as food packaging, and at end-of-life their carbon atoms can be recovered for soil enrichment after composting, so that new food or materials can be produced. Packaging functionality, such as mechanical, gas barrier, and heat-seal performance, of emerging biodegradable packaging, with a laminated, coated, monomaterial, and/or blended structure, is not yet well known in the food industry. This lack of knowledge, in addition to end-of-life concerns, high cost, and production limits is one of the main bottlenecks for broad implementation in the food industry. This study determines application areas of 10 films with a pragmatic approach based on an experimental broad characterization of packaging functionality. As a conclusion, the potential application of these materials is discussed with respect to industrial settings and food and consumer requirements, to support the implementation of commercially available, biodegradable, and, more specifically, compostable, materials for the identified food applications.

## 1. Introduction

Plastic materials have been increasingly applied in packaging over the last several decades because of their low cost, low weight, and customizable functional properties. In 2019, 368 million tons of plastics were produced globally, of which a staggering amount of around 40% is used in packaging [1]. To reduce the amount of plastic waste, global and local initiatives, such as the European directive (EU) 2018/852 [2], are established, that fit into a vision of a circular economy of plastics. The circular economy diagram of the Ellen MacArthur foundation illustrates a continuous flow of technical and biological materials through the value circle [3].

Plastic biodegradation is defined as the microbial conversion of all its organic constituents to carbon dioxide (CO_2_), new microbial biomass, and mineral salts under aerobic conditions [4]. Composting of biodegradable packaging is described in the DIN EN 13,432 standard [5]. Besides composting, anaerobic degradation systems that produce methane gas are emerging. Currently, only a small fraction of globally produced plastics is biodegradable (1.553 million tons in 2021), but this amount is predicted to rise to 5.297 million tons in 2026 [6]. With a low but increasing availability of biodegradable plastics, this group of materials can become an emerging alternative to mechanical recycling and reuse in a long-term organic circular economy. Packaging is already the main application of biodegradable plastics, with 43% and 16% of biodegradable materials being applied as flexible and rigid packaging, respectively [4].

With a projected growth from $338 billion in 2021 to $478 billion in 2028, the food packaging market plays an important role in our society [7]. Considering the number of food packages, plastic and paper are the most important materials for food applications [8]. In food packaging films, different materials are often combined to obtain high-performing and cost-effective packages. This can be done by blending, coating, or laminating. In order to maintain biodegradability by composting, it is important that these composites are made of compostable materials. However, small fractions of non-compostable materials, limited to a maximum content of 10% because of degradation and disintegration criteria, can be allowed for composting if the whole package meets the demands of the DIN EN 13432 standard [4]. Industrial and home compostability can be differentiated; these processes differ in temperature and time. Polylactic acid (PLA), polybutylene adipate terephthalate (PBAT), polybutylene succinate (=PBS), and polyhydroxyalkanoates (PHAs) are biodegradable plastics that were subjects of previous studies on packaging functionality in food applications [9,10,11]. These materials are industrial-compostable [12]. Depending on the properties of the coating, coated paper can be considered as biodegradable packaging. Interest is increasing for its implementation in food packaging, mainly because of the versatile end-of-life options of this material [13,14]. Cellulose, the main component of paper, is a natural polymer that can be easily obtained from the cell walls of plants. Processes to extract and modify cellulose are subjects of recent studies, of which the lyocell process is one example [15]. Plant waste streams can be valorized by extracting cellulose to make packaging films. A recent study extracted cellulose from cocoa pod husk, a waste stream of the chocolate industry, to develop biodegradable cellulose films [16]. Cellulose and its derivatives can be found in food packaging films, such as solution-casted cellulose acetate, extruded cellulose nanocrystals, electrospun hydroxymethyl cellulose, and many others [17]. Starch is another example of a natural abundant polymer that can be used in packaging. This polymer is home-compostable, which is a less aggressive process than industrial composting. Also, cellulose is home-compostable, if the lignin content does not exceed a threshold value of 5% [12].

In a 2021 survey, among 24 European food companies and packaging material providers, functionality of biodegradable materials was indicated, in addition to high cost, low availability, and end-of-life concerns, as a bottleneck for implementation in food packaging. Because of the interest of the food industry in packaging functionality of biodegradable materials, the research project BIOFUN evaluates typical food packaging functionalities, such as mechanical, gas barrier, and heat-seal performance of commercially available films in 2021 and 2022 [18].

The objective of this study is to determine application areas in food packaging of currently commercially available biodegradable films. A pragmatic approach is followed, based on a broad characterization of the mechanical, seal, and gas barrier performance. Additionally, opacity and water contact angle are determined for further characterization.

## 2. Materials and Methods

### 2.1. Materials

Table 1 lists 10 films that were supplied by companies participating in the BIOFUN project. Results of thickness measurements and the main components of the seal side, identified with attenuated total reflectance Fourier transform infrared (ATR-FTIR) spectroscopy (spectra are not shown), are added to this table to give supporting information of these samples. The identified components with FTIR compensate for the lack of commercially available information, which is the result of the high level of secrecy on the chemical composition in the industry. The list includes paper, PLA, PBAT, PBS, poly(butylene succinate-co-butylene adipate) (PBSA), starch, cellulose and poly(3-hydroxybutyrate-co-3-hydroxyvalerate) (PHBV), which are considered for use as food packaging. Coated paper 1, with PE as coating material, is unlikely to be compostable. The materials of Table 1 are differentiated in four material groups: coated papers, cellulose films, pilot extrusions and commercial monolayers. Two coated papers, two cellulose films, two rather thick pilot extrusions and four commercial monolayers, subdivided in two monolayer monomaterials and two monolayer blends, are the subjects of this study. Results of materials in each group are mutually compared and discussed. Digital photos of the samples in Table 1 are shown in Figure 1.

### 2.2. Methods

To compare the test materials based on their packaging performance, the mechanical, gas barrier and seal characteristics are determined for all samples. Tests are performed by machine direction in a standard climate (23 °C, 50% relative humidity (RH)), unless otherwise stated. Standard deviations are calculated to show the level of scattering of results.

#### 2.2.1. Mechanical Performance

Thickness is tenfold-measured according to ISO 4593. Peak stress (N mm^−2^) and total strain (%) are determined in fivefold measure with a tensile tester. Dumbbell-shaped samples with 3.18 mm width of the narrow section, described in ASTM D638 [19], are used to prevent the samples from breaking at the clamp. Total strain values are mainly used for mutual comparison. No extensometer is used, so comparisons of total strain values in the literature must be made with caution. Slipping is prevented by clamping the wide section in diamond-coated jaws. A clamp distance of 20 mm and a separation rate of 100 mm min^−1^ are used to perform the test. Additional tests in a temperature chamber are done to evaluate the impact of environmental temperature on peak stress and total strain. Relevant temperatures for food processing, ranging from freezing at −18 °C until pasteurization, hot fill and/or microwave at 100 °C and/or melting of the sample, were considered in this test.

Maximum force (N), total displacement (mm), and total energy (mJ) are determined in fivefold with a puncture-resistance test. A penetration probe, as described in ASTM F1306 [20], moves toward the outer side of a clamped film with a speed of 25 mm min^−1^ until the film is penetrated.

Tear resistance (mN) is determined in tenfold with an Elmendorf test, which uses a pendulum to propagate an existing slit, as described in ISO 6383-2 [21].

#### 2.2.2. Gas Permeability

Single measurements are done in standard conditions to screen the oxygen transmission rates (OTR) of all samples at 23 °C and 0% relative humidity, as described in ASTM F1307 [22]. Additional tests on high gas barrier materials are performed at 23 °C and 50% relative humidity, following ASTM F1927 [23], at both sides of the film.

Single measurements are done in extreme test conditions to screen the water vapor transmission rates (WVTR) of all samples in a worst-case scenario. WVTR, according to ASTM F1249 [24], is determined at 38 °C and 100% relative humidity at the outer side of the film, while 0% relative humidity is maintained at the inner side.

#### 2.2.3. Seal Performance

Seal temperature is varied with two hot jaws, at a seal time of 1.0 s and a seal pressure of 1.0 N mm^−2^. Samples of 30 mm width are sealed while Teflon sheets are used on both sides to prevent the material from sticking against the jaws. At each temperature, three samples are sealed. Seal strength, following ASTM F88 [25], is evaluated in a timeframe of 4 h after sealing. Then 15-mm-wide samples are clamped with a distance of 20 mm and separated at a rate of 300 mm min^−1^. Three characteristics of the sigmoidal seal curve are determined: an initiation temperature, which is the jaw temperature at which seal strength exceeds a threshold value of 0.05 N mm^−1^ [26]; a mid-slope temperature, which is the jaw temperature at which half of the maximum seal strength is exceeded; and the maximum seal strength.

Hot tack tests, following ASTM F1921 [27], are performed on 15-mm-wide samples at a test speed of 200 mm s^−1^. Seal time and seal pressure are respectively set at 1.0 s and 1.0 N mm^−2^, while the seal temperature of two Teflon-coated hot jaws is varied. At each temperature, three samples are measured. Seals are evaluated 0.1 s after opening of the seal jaws. Four characteristics are determined: seal initiation temperature, which is the jaw temperature at which a threshold value of 0.03 N mm^−1^ is exceeded [28]; the temperature of maximum strength, which is the jaw temperature at which hot tack strength reaches its maximum; the hot tack window, which is the temperature range of the jaws where hot tack strength is higher than 0.1 N mm^−1^ [28]; and the maximum hot tack strength.

In addition to the above-described broad seal characterization, additional seal tests can be done to check the compatibility with specific food applications. This is done with real food contamination in seal through contamination tests. Two case studies, which relate film samples with food applications, are defined based on gas barrier performance. Low gas barrier samples are evaluated with contamination types that are related with unprocessed fruit and vegetables. In this application, water droplets and solid soil particles are expected. Sand and coffee particles are selected as simulants of soil particles. High gas barrier samples are evaluated as grated cheese packaging. Square samples of approximately 10 cm × 10 cm are cut and attached to a cardboard tool with plastic tape. A rectangle of 20 mm × 40 mm was marked in the center of the sample to ensure that the contamination was distributed over the entire length of the seal. Then, 10 mg of the solid contamination or 30 µL of water was evenly spread into the rectangle to maintain a 12.5 g/m^2^ or 37.5 mL/m^2^ contamination density. Specifically, for grated cheese, three strings were placed vertically and distributed in the middle and the two corners of the rectangle. A second sample was also attached to the cardboard tool to cover the contamination. In a final step, the tool was manually placed between the hot bars, forming the seal. The above-described set-up is illustrated in Figure 2.

In a previous study, solid contamination was applied in a standardized method and seal through contamination performance was evaluated with a design-of-experiments (DOE) approach [28]. This approach was followed, with the exception of adding contamination as a categorical parameter in the design space. For the low gas barrier samples, three levels are considered for seal temperature, time, and pressure, and contamination was added as a categorical variable with four levels: clean, ground coffee, sand, and water. Three replicates are carried out for each contamination level. Main-order, second-order and interaction effects are included with seal strength as response, resulting in 41 runs. A similar approach is followed for the contamination experiments with the high-barrier cellulose samples, with the exception of only two considered levels for contamination: clean and grated cheese, resulting in 24 runs. After experimentation, a standard least square method is followed to fit a model. Second-order and interaction terms with a *p*-value above 0.05 were not used in the model. Seal strength is maximized for clean seals, and the predicted values are validated by performing five measurements at maximal settings. All contaminations are also validated at equal settings to allow comparison between clean and contaminated seal strength. For more details on this approach, the reader is referred to the previous study [28].

#### 2.2.4. Additional Characterization

Opacity is measured to show the appearance and decoration potential in food packaging. The Hunter lab method in the reflectance mode is followed. The opacity Y (in %) is calculated by dividing the opacity on a black standard Y_b_ with the opacity on a white standard Y_w_. For each sample, average values of four measurements, twice on each side, are calculated.

Water contact-angle measurements are carried out to characterize hydrophobic properties of the samples. Samples are cut to fit the sampling area. A 2-µL MQ water (18.2 MOhm cm) drop is gently deposited on the seal surface by using a micro-syringe, and digitally photographed immediately. Contact angles are measured at both sides. Average values of contact angles of 15 drops at different spots on the surface of each sample are calculated.

#### 2.2.5. Apparatus

Thickness is measured with a precision thickness gauging model 2010 U (Wolf Messtechnik GmbH, Freiberg, Germany). Tensile, puncture, and seal-strength tests are performed with a 5ST universal testing machine (Tinius Olsen Ltd., Redhill, United Kingdom), inside a TH 2700 temperature chamber (Thümler GmbH, Nürnberg, Germany). Tear resistance is tested with a tearing tester ED 300 (MTS Adamel Lhomargy, Roissy-en-Brie, France). Dry and humid oxygen permeation are respectively measured with the OX-TRAN^®^ model 702 and the OX-TRAN^®^ model 2/21 SH (Ametek Mocon, Brooklyn Park, MN, USA). Water vapor permeation is measured with the Permatran-W models 3/33 MG and SW (Ametek Mocon, Brooklyn Park, MN, USA). Seals are prepared with a Labthink HST-H3 heat seal tester (Labthink Instruments Co Ltd., Jinan, China). Hot tack samples are evaluated with a J&B Hot Tack Tester model 5000 MB (Vived-Management, Lanaken, Belgium). Opacity is measured with a Datacolor Check3 (Datacolor België BVBA, Ghent, Belgium). Water contact angle is measured with a GBX Digidrop contact angle (GBX Scientific, Dublin, Ireland).

## 3. Results

### 3.1. Mechanical Performance

Table 2 shows the average values and standard deviations of the mechanical characterization of all materials in standard climate (23 °C, 50% RH). Representative stress–strain curves of each of the samples are shown in Figure 3.

Because of the high relevance of processing temperatures in the food industry, such as in freezing, cooling, hot filling, microwaving and/or pasteurizing, environmental temperature is varied in tensile tests of a selection of materials. Thin commercial films with no backing layer, with the addition of coated paper 2 and cellulose 1, are evaluated in this test. Samples are tested at −18, 4, 23, 40, 60, 80, and 100 °C. The results of peak stress and total strain are shown in Figure 4 and Figure 5.

### 3.2. Gas Permeability

Table 3 shows the transmission rates for oxygen gas and water vapor.

### 3.3. Seal Performance

Table 4 shows the results of the seal characterization.

Two cases are studied in additional seal experiments with contamination: coated papers 1 and 2, with relative low gas barriers, for unprocessed fruit and vegetables, and cellulose films, with relative high gas barriers, for grated cheese. Table 5 shows the predicted maximum seal-strength values for clean and contaminated seals of all cases at optimal seal parameters.

### 3.4. Additional Characterization

The results of opacity and water contact angle are respectively shown in Table 6 and Table 7.

## 4. Discussion

### 4.1. Mechanical Performance

Coated paper shows moderate peak stress values in Table 2. As a result, actual tensile forces will be high because of the rather thick materials that are used in food packaging. The strain of coated paper is limited because of the immediate break of the paper substrate in a tensile test, high variations can be caused by delamination of the plastic coating. In a previous study on PLA coated paper, tensile stress and elongations ranged, respectively, from 58–75 N mm^−2^ and 3–4% [29]. However, paper type, coating material, and coating thickness impact, among others things, the mechanical properties of coated papers. The puncture results show moderate forces, small displacements, and moderate energies. Moreover, tear resistance was moderate, compared to other samples.

Cellulose 1 was the strongest material in the tensile test. The decreased peak stress of cellulose 2 is probably caused by the lamination with a weaker but tougher PBS layer. Cellulose films have limited strain because of the almost immediate break of the brittle cellulose layer in a tensile test, and high variations of cellulose 2 are caused by the delamination of the tough seal layer. The experimental values of stress and strain of cellulose 1 are equal with values in the datasheet of commercial cellulose film [30]. Puncture resistance forces and energies of cellulose films are high, and displacements are moderate. The tear resistance of cellulose 1 reaches the lowest value of all samples. This property can be dramatically improved by laminating a tough seal layer, as observed in the results of cellulose 2, which has a laminated PBS layer.

The pilot extrusion of PBS is mechanically superior to that of PHBV, with the exception of tear resistance. There is no comparable value found in the literature for the PHBV–PBAT blend. In a review on monomaterial PHBV [11], a tensile stress range of 18–45 N mm^−2^ is found. The peak stress value of the PHBV film of this study, which is blended with PBAT, mineral filler and process additives, fits within the range of monomaterial PHBV. PBS is strong and tough at the same time, and this is reflected in the tensile and puncture results. In the comparison of the puncture and tear resistance results of the pilot extrusions with the commercial films, caution must be taken with puncture and tear resistance, because of the different thickness.

The strong mechanical performance of PBS is also reflected in the results of the commercial monolayer, reaching a moderate peak stress and very high strain in the tensile test. The stress values of the two PBS-based films, the pilot extrusion and the commercial monolayer, are relatively high, compared to the stress values, ranging from 20 to 34 N mm^−2^, found in a study on poultry meat packaging [10], bread packaging [31], and a recent review on PBS properties [32]. The increase in strength of the films in this study indicate a difference in production, which is known of the pilot extrusion film, by blending with PBSA and process additives, but is not known of the commercial monolayer. A previous study on PBS blends showed that mechanical properties were majorly influenced by compatibility between polymers and morphology, including microstructures and crystallinity [31]. A moderate puncture force, high displacement and high energy in the puncture test are achieved. Tear resistance of PBS is rather low compared to other samples. PLA also stands out as a mechanical good performing film, with high peak stress and moderate strain in the tensile test, and high force, displacement, and energy in the puncture test. Also, this film is easy-tearable. The two blended films with PBAT are characterized with low strength, high toughness, and very high tear resistance. A previous study on the mechanical properties of PLA and PLA-PBAT blended films illustrates a strong but brittle tensile performance of PLA film and a weaker but tougher performance of the PLA-PBAT blended film [33]. PBAT is often used in blends to increase flexibility and toughness of brittle biodegradable materials.

The results in Figure 4 and Figure 5 are discussed below.

With the exception of cellulose 1, peak stress tends to decrease at increasing temperatures. The tendency for total strain is less clear. PBS, PLA, and the PBAT blends could not be tested at high temperatures because of high stickiness. With respective glass transition and melting temperature values of PBS and PLA of −32 °C and 114; 59 °C and 154 °C, it is clear that the sticky behavior occurs above glass transition temperature [34].

The peak stress of coated paper 2 decreased from 51 N mm^−2^ at −18 °C to 23 N mm^−2^ at 100 °C while remaining brittle at all environmental temperatures of Figure 4. The deviating results of total strain at 4 and 23 °C were caused by delamination of the plastic coating. Cellulose 1 remains very strong, mostly above 100 N mm^−2^, and brittle, with strain values ranging from 8 to 22%, at all considered temperatures.

PBS remains strong up to 60 °C. The total strain decreased below 100% at cool temperatures.

PLA showed a bigger temperature-dependent peak stress behavior, compared to PBS, achieving 89 N mm^−2^ at −18 °C and 22 N mm^−2^ at 80 °C. The drop in tensile stress from 20 to 60 °C is previously illustrated in another study on the mechanical performance of PLA tensile specimens, attributed to approaching the glass transition region of PLA [35]. The total strain decreased below 20% at cool temperatures.

PBAT blends have low peak stress values, between 24 and 40 N mm^−2^ at cool temperatures and 12 N mm^−2^ at 60 °C, but high total strain values.

In conclusion, the mechanical characterization of coated paper and cellulose-based films can be described as strong but very brittle materials. The low strain values, compared to other tougher samples, are illustrated in Figure 3. However, brittleness might be overcome by laminating a tough layer. Both materials can be used over a wide temperature range, from freezing at −18 °C up to 100 °C. The film with PHBV is rather weak and brittle, compared to the other materials. The films with PBS and PLA are strong and tough materials under standard conditions. The toughness, however, decreases at low temperatures. On top of that, stickiness initiates well below 100 °C, which will restrict their use to a narrow temperature range, especially if moderate toughness is required. If brittleness is not a big issue, these materials can be used in cold and standard temperatures. The blended PBAT films, with starch or PLA, are rather weak but very tough, even at cool temperatures. Because of the melt initiation well below 100 °C, the use of these blends is restricted to cold and standard temperatures.

### 4.2. Gas Permeability

In Table 3, coated paper 1 shows similar barrier properties as polyolefin film, because of its high OTR and rather low WVTR values [36], pp. 259–308. This gas barrier performance can be related with the presence of low-density polyethylene (LDPE) at the seal surface, identified with ATR-FTIR. A 25-µm pure LDPE reference film has OTR between 6500 and 7800 cc m^−2^ d^−1^, measured at 23 °C and 0% RH, and WVTR between 12 and 19 g/m^2^.d, measured at 38 °C and 90% RH [36], pp. 259–308. The values of coated paper 1 correspond with TR values of 10–15 µm LDPE. Coated paper 2 on the other hand is a low gas barrier material for food packaging applications. Specifically, for WVTR of coated paper, a recent study compared high gas barrier coated papers at 23 °C, 85% RH and 38 °C, 85% RH and suggested that the integrity of the barrier layer was disrupted at 38 °C [37]. The authors of that study suggest using milder test conditions to simulate more closely the environment of food packages and to prevent disruption of barrier layers.

Because of the low OTR values of the cellulose films, additional oxygen measurements at 50% RH are done to check the influence of humidity on oxygen transmission. With respective values of 3.7 and 5.8 cc m^−2^ d^−1^, it is clear that the OTR increases with increasing RH. These cellulose films have barrier coatings because neat cellulose is a low gas barrier for food applications. Both films achieve values similar to poly(vinylidene dichloride) (PVDC)-coated materials. PVDC, which is a high gas barrier for food applications [36], pp. 259–308, is identified in the seal surface with ATR-FTIR in cellulose 1, but not in cellulose 2. Cellulose 2 is, however, laminated with a PBS layer that obstructs identification with ATR-FTIR of parent layers. These films can be used to maintain modified atmosphere in food packages.

Paper and cellulose are low-barrier substrates that require a barrier layer, such as in coatings, to improve the barrier properties. This is illustrated in Figure 6. Barrier properties of such coated materials are mostly attributed to thin barrier layer(s) in the coating. Coating thickness, multilayer architecture, individual layer composition and concentration gradient are determining factors in this process [36], pp. 259–308. An example of such a process is the transmission of water vapor in the atmosphere, across a packaging material, in the dry headspace of food applications, such as cookies. In some applications, such as yogurt, the process is reversed.

A previous study, which produced biodegradable, blown extruded films of blends of thermoplastic starch and PBAT, functionalized with plasticized nitrite, measured a relatively low oxygen permeability with a permeability coefficient down to 1.2 cc mm m^−2^ d^−1^ for films with 5% nitrite content [38]. This coefficient corresponds with an OTR-value of 24 cc m^−2^ d^−1^, considering a film of 50 µm thickness. There is still a gap between this moderate value and those that are measured with the commercial cellulose films in this study. More research is needed to obtain biodegradable food packaging with the permeation levels of the cellulose films in this study, without the need of non-biodegradable functional components.

The pilot extrusions and monolayer films are low gas barrier materials for food packaging applications. The application of low gas barrier samples, such as the coated papers, the pilot extrusions, and the monolayer films, is restricted to foods with low-barrier or high-respiration requirements such as unprocessed fruit and vegetables with short shelf lives. With these food applications, the high permeation of water vapor and oxygen gas is required to avoid, respectively, the accumulation of saturated water vapor which leads to fungal growth, and an anoxic condition [39]. If a high gas barrier is required, these films need to be coated and/or laminated with materials that are able to add this property.

Coated paper 1 might be used for applications that need a water vapor barrier, but no oxygen barrier, which can be the case for some dry foods, such as flour, dried pastas, crackers, and cookies. The barrier cellulose films can be used for applications with oxygen and water vapor barrier requirements. Typical examples are cheese, meat, high-fat products, and ready meals [36], pp. 259–308.

### 4.3. Seal Performance

Seal strength, hot tack strength initiation, and mid-slope temperatures, in Table 4, are, with the exception of the thick pilot extrusion films, below or equal to that of typical polyolefin-based seal layers, such as LDPE, ionomers, or metallocene plastomers [36], pp. 181–257. Six out of ten films achieve over half of the maximum seal strength at jaw temperatures below 100 °C. These materials can be considered in high-speed packaging operations.

Because uncoated paper cannot be heat-sealed, heat-seal characteristics of coated paper are mainly attributed to the coating material, coating thickness, and coating process. Coated paper 2 outperforms coated paper 1, with lower initiation temperatures and higher hot tack strength. It is capable of maintaining a minimum hot tack strength threshold value of 0.1 N mm^−1^ over a very wide temperature region of 110 °C. The seals of coated paper fail by delamination of paper fibers during seal strength and hot tack tests.

Cellulose 2 has lower initiation temperatures and higher strengths than cellulose 1. The better seal performance of cellulose 2 is attributed to the lamination of a PBS layer with excellent seal properties. The seals of cellulose 1 fail by peeling cohesively, whereas those of cellulose 2 fail by breaking unsealed material during a seal strength test. The difference in the failure mechanism is related with the big difference in maximum seal strength. In the hot tack test, both materials fail by peeling cohesive. The different seal failure mechanism of cellulose 2 in the hot tack test, compared with the seal strength test, is related to the very low cool time. The seal is evaluated 0.1 s after opening of the hot jaws, when it is still hot.

The pilot extrusion films show high initiation temperatures; this is typical with heat conductively sealed thick films, where heat is transferred through a thick layer, from the hot jaws to the outer layers and the seal interface so entanglement can occur. The seals of the PHBV blend fail by peeling cohesively, whereas those of the PBS blend fail by breaking unsealed material during a seal-strength test. In the hot tack test, a break in the proximity of the seal is observed with both materials. The presence of a weak spot in the remote materials is suggested as a hypothesis. The weak spot is still hot, but thinner than the seal area. Both thick pilot extrusion films can be heat-sealed, but a thinner commercial structure should be evaluated to determine specific application areas for these materials. The thin PBS and PLA monolayers have low initiation temperatures and rather high strengths for materials without rigid backing layers. The seals of these materials fail by breaking unsealed material during a seal-strength test. In the hot tack test, both materials peel cohesively and/or break in the proximity of the seal. PLA has the advantage of maintaining its hot tack strength over a wide temperature range. The thin monolayers with PBAT also seal at low temperatures but strengths are rather low. Both PBAT blends show similar seal failure mechanisms than those observed with the PLA and PBS monolayers. Low seal strengths are beneficial in easy-peel applications. In a previous study, that evaluated the seal performance of several PLA-PBAT blend ratios, sealed to a PLA container, the blended films were characterized as easy-peel [9].

It can be concluded that coated paper 2, cellulose 2, and PLA are very well-suited for packaging operations where the hot seal is put under pressure, such as in vertical-form-fill-sealing or when springback forces are induced, immediately after sealing, for example, by solid food contaminants in the seal area. The thin PBS monolayer could similarly be used, but a stricter temperature control is advised because of the smaller hot tack temperature window. The use of coated paper 1 is restricted to operations where the hot seal is not pressurized. Cellulose 1 and the two thin PBAT blends are heat sealable, but their use is restricted to applications where low strength is required, such as in packaging of low-weight foods or easy-peel applications.

The optimal parameters in Table 5 are equal for clean and contaminated seals because all interaction terms of contamination with a seal parameter are not significant and are left out in the fitted models. The results of individual runs, coefficients, and *p*-values of terms in fitted models are not shown because of the sole objective on evaluation of the clean and contaminated maximal seal strengths. A 95% confidence interval is calculated, based on 5 experiments at optimal seal parameters. Only with clean-coated paper 1, water-contaminated coated paper 2 and grated cheese contaminate cellulose 2, predicted values are slightly outside the confidence interval. All other predicted maxima fall in a 95% confidence interval.

All considered materials have overlapping confidence intervals for clean and contaminated seals, so the clean maximal seal strengths can be matched with contamination. Powder contamination densities of 12 g m^−2^ and above are related with aggregate formation and a decrease in maximum seal strength of polyethylene film [40]. For the considered coated papers, this threshold value can be exceeded while the maximum seal strength is maintained. Further experiments with higher contamination densities can be done to study the limits for these materials. Both coated papers can be considered to pack fresh foods. Further experimental tests and/or finite element analysis with target foods and packaging with specified dimensions can be done to check if the seal strength of these coated papers is sufficient for the food packaging application. The barrier cellulose films can be considered when packing grated cheese. The very low seal strengths of cellulose 1 makes this material not suited for heavy weight applications. One might think of combining the good seal through contamination performance, almost equally strong hot tack, shown in Table 4 and easy-tear features, shown in Table 2, of cellulose 1 in easy-tearable low-weight packages. Cellulose 2 can be used in packages with higher weight in cheese. Besides additional mechanical analysis of the entire food packaging concept, to check if seal strength is sufficient, additional leak tests are advised, because of the importance of good barrier properties of grated cheese packaging.

### 4.4. Additional Characterization

Opacity, which is normalized to thickness with homogeneous film structures in previous studies, is correlated with film thickness [41,42]. Besides thickness, variations in opacity can be related with the material composition, such as the reflection of light of foreign nanoparticles [43]. There is also an obvious impact of printing and coloration on opacity. The opacity results in Table 6 show big differences between the samples. Non-transparent samples, as shown in in Figure 1, such as the coated papers and the black PLA + PBAT blend, have high opacity values. Food packaging with transparency properties are, however, preferred by consumers [44]. Samples with low opacity values, such as PLA and cellulose 1 approach full transparency, with respective values of 7.9 and 11.5. These values are in the same range as other biodegradable films that were measured with the same method [43]. Other thin samples have hazier appearances, which is reflected by increased opacity values. The thicker pilot extrusions have moderate opacity values compared to other samples.

The tendency of food to adhere to the packaging surface determines to a large extent the preservation of food [45]. Hydrophobic properties of the surface are desired to improve the resistance of chemical interactions with food by minimizing the contact area. The values in Table 7 are in a narrow range of 80–105°, between that of smooth cellulose films, which are hydrophilic and have contact angles below 50°, and superhydrophobic surfaces, a property that can also be achieved with biodegradable materials, characterized by contact angles above 150° [46]. The standard deviations of the results are rather high, suggesting inhomogeneous surfaces, compared to reported values in the literature [47,48,49]. The water contact angle of coated paper 1 is similar to a value of LDPE, reported in a previous study [47]. Contact angles of PBAT blends, with thermoplastic starch and nano zinc oxide, in a previous study were in between 89 and 104° [49]. This range is similar to the ranges of the values of the PBAT blends on the surface of the samples in this study, such as coated paper 2, pilot extrusion PHBV, and the two monolayer blends, PLA + PBAT and starch + PBAT. Another study reports a low value of 57° for PBAT [47], which highlights the difficulties comparing these values in the literature. The same study reports a value of 68° for PLA, whereas the value for PLA in this study is 80°, which is low compared to the other samples. The two PBS samples of this study are with values of 84° for the thin monolayer and 104° for the pilot extrusion, also higher than a value reported in a previous study [48]. In conclusion, water contact-angle values of the samples in this study are higher than or equal to values found in the literature. This is probably related with modifications in commercial food packaging films, in order to decrease the contact area with food.

## 5. Conclusions

Coated papers and high-barrier cellulose films are brittle materials with a potential use over a wide environmental temperature range. Barrier and/or heat seal properties can be altered with the appropriate plastic coating. The case studies to check the seal through contamination performance show that maximal seal strength can be maintained.

In a comparison of two thick pilot extruded films, the PBS blend is stronger and tougher than the PHBV blend at standard environmental temperature. Without the use of additional gas barrier layers, application of these materials is restricted to food with low- barrier requirements, such as takeaway meals and unprocessed fruit and vegetables. Both materials can be heat-sealed. In order to be able to determine seal application areas, film production need to be optimized to obtain commercial structures, such as thin flexible films or trays.

The application of PBS, PLA, a PLA-PBAT blend, and a starch-PBAT blend is restricted to food with low-barrier requirements. Additional barrier layers, of which the identified PVDC layer in high gas barrier cellulose film is an example, are needed to implement these materials for food with high-barrier requirements, such as meat, cheese, high-fat products, and ready meals. Monolayers with PLA and PBS combine high strength and toughness at standard environmental temperatures. However, the temperature window of these good mechanical features is narrow. Both materials are able to produce strong seals with low initiation temperatures. Both materials can be applied as strong seal layers in high-speed VFFS applications or as heavy-duty monolayers in standard environmental temperatures. The application at cold temperatures can be considered if the low maximum strains are sufficient for the specific food packaging. The PBAT blends are weak but tough from cold to standard environmental temperatures. Application is restricted at temperatures above 60 °C. These materials can be applied as relatively weak seal layers, which is of high interest in easy-peel applications, and as light-duty monolayer in cold and standard environmental temperatures.

Depending on the selection of coated and/or laminated materials, the application potential of biodegradable materials in food packaging is very broad, ranging from low barrier packaging of low-weight foods at standard temperature, to high-barrier packaging, such as modified atmosphere packaging of high-weight foods, extreme temperature processing and/or high-speed applications, such as the vertical form fill seal. Biodegradable food packaging is emerging. This study fully supports the implementation of commercially available biodegradable materials for the identified food applications.

## Figures and Tables

**Figure 1 polymers-14-02569-f001:**
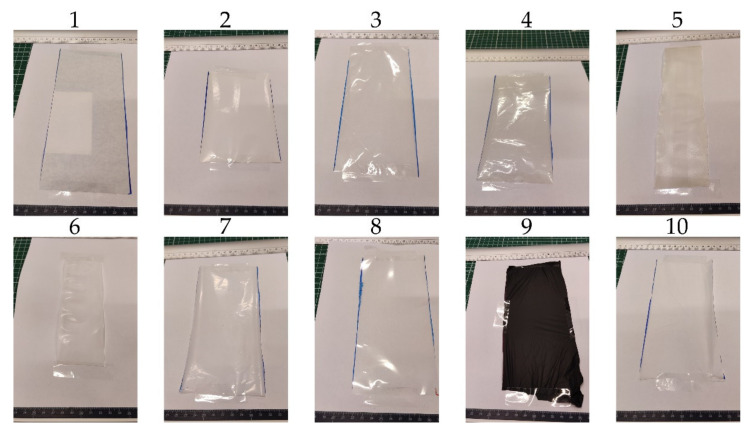
Digital photos of samples on white paper. (1: Coated paper 1; 2: Coated paper 2; 3: Cellulose 1; 4: Cellulose 2; 5: Pilot extrusion PHBV; 6: Pilot extrusion PBS; 7: PBS; 8: PLA; 9: PLA+PBAT; 10: Starch + PBAT).

**Figure 2 polymers-14-02569-f002:**
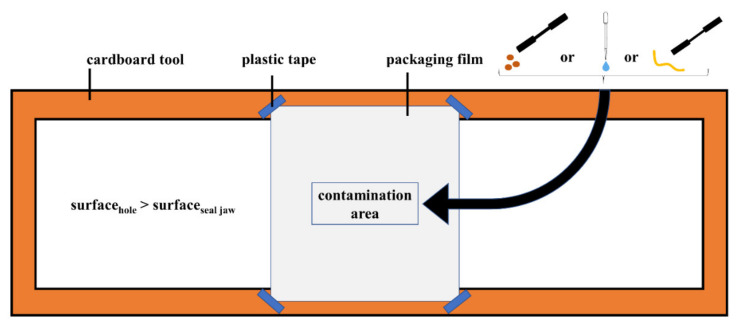
Set-up to contaminate the seal area.

**Figure 3 polymers-14-02569-f003:**
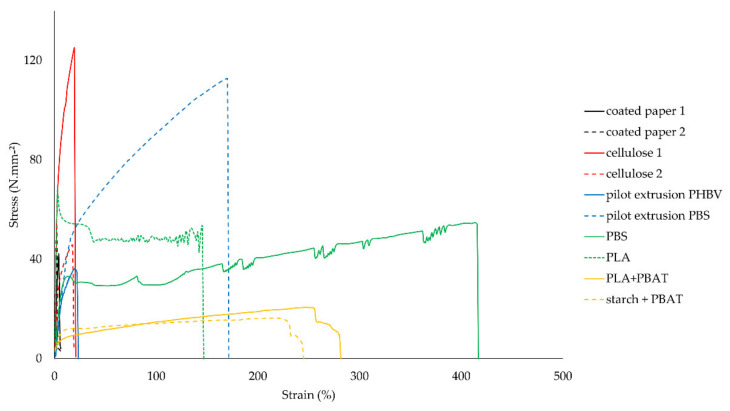
Stress−strain curves.

**Figure 4 polymers-14-02569-f004:**
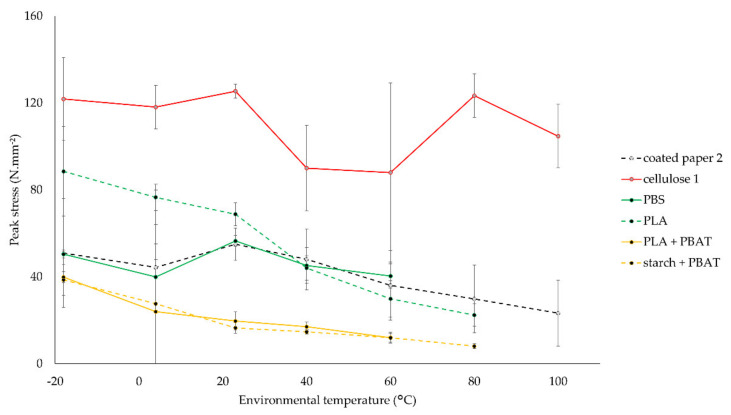
Impact of environmental temperature on average values of peak stress of biodegradable films and standard deviations (n = 5).

**Figure 5 polymers-14-02569-f005:**
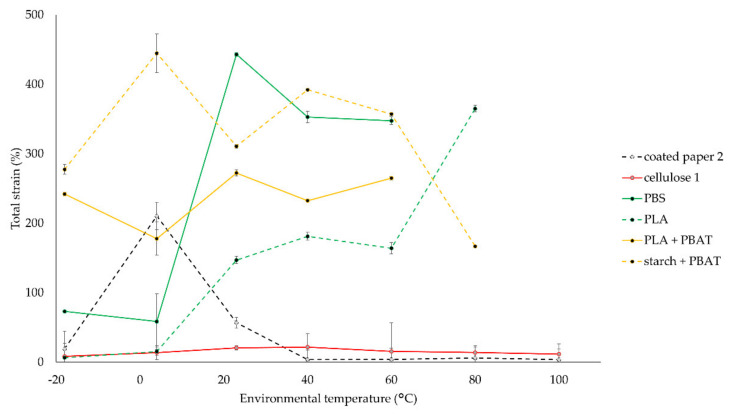
Impact of environmental temperature on average values of total strain of biodegradable films and standard deviations (n = 5).

**Figure 6 polymers-14-02569-f006:**
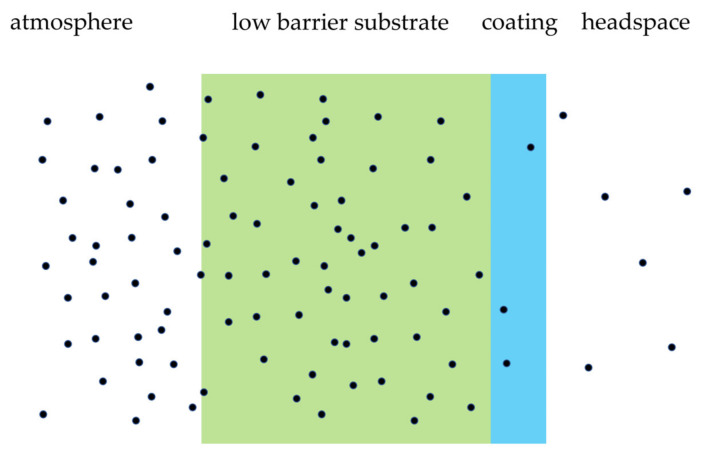
Permeation of gas and/or vapor, from atmosphere to headspace, through coated, low-barrier substrates.

**Table 1 polymers-14-02569-t001:** Sample description.

Code: Details	Thickness (mm) (n = 10)	Identified Components of Seal Surface ^1^
**1. Coated paper 1**: commercial coated paper	0.097 ± 0.003	LDPE
**2. Coated paper 2**: commercial coated paper	0.076 ± 0.002	PLA, PBAT
**3. Cellulose 1**: commercial coated cellulose film	0.030 ± 0.000	Cellulose, PVDC
**4. Cellulose 2**: commercial laminated cellulose film	0.087 ± 0.002	PBS
**5. Pilot extrusion PHBV**: monolayer blend of PHBV + PBAT + mineral filler + process additives	0.264 ± 0.005	PHBV, PBAT
**6. Pilot extrusion PBS**: monolayer blend of PBS + PBSA + process additives	0.284 ± 0.002	PBS, PBSA
**7. PBS**: commercial monolayer	0.047 ± 0.001	PBS
**8. PLA**: commercial monolayer	0.030 ± 0.001	PLA
**9. PLA + PBAT**: commercial monolayer blend	0.020 ± 0.001	PBAT, PLA, CaCO_3_
**10. Starch + PBAT**: commercial monolayer blend	0.025 ± 0.003	PBAT

^1^ Identified with ATR-FTIR.

**Table 2 polymers-14-02569-t002:** Results of mechanical characterization.

	Tensile		Puncture			Tear Resistance
Samples	Peak Stress ^1^ (N mm^−2^)	Total Strain ^1^ (%)	Max. Force ^2^ (N)	Total Displacement ^2^ (mm)	Total Energy ^2^ (mJ)	Tear Resistance ^3^ (mN)
1. Coated paper 1	37.6 ± 6.1	5.28 ± 0.49	12.2 ± 1.5	2.91 ± 0.13	16.4 ± 2.0	663 ± 37
2. Coated paper 2	55.1 ± 7.5	56.8 ± 72.8	7.35 ± 0.82	2.95 ± 0.21	11.3 ± 0.7	455 ± 41
3. Cellulose 1	125 ± 3	20.7 ± 1.5	16.7 ± 1.0	5.03 ± 0.34	36.4 ± 4.3	76 ± 4
4. Cellulose 2	46.5 ± 2.4	199 ± 244	17.1 ± 0.9	4.79 ± 0.18	34.4 ± 2.9	680 ± 104
5. Pilot extrusion PHBV	37.8 ± 1.8	24.9 ± 2.8	8.62 ± 0.65	3.77 ± 0.07	20.5 ± 1.7	526 ± 40
6. Pilot extrusion PBS	106 ± 5.0	165 ± 17	54.6 ± 1.0	7.43 ± 0.29	194 ± 10	375 ± 19
7. PBS	56.5 ± 2.6	443 ± 22	10.3 ± 0.6	8.65 ± 0.36	57.2 ± 5.5	127 ± 67
8. PLA	68.8 ± 5.4	147 ± 29	13.4 ± 1.9	7.85 ± 1.07	59.3 ± 16.5	142 ± 4
9. PLA + PBAT	19.7 ± 4.2	272 ± 44	1.28 ± 0.08	6.65 ± 0.34	6.03 ± 0.56	992 ± 189
10. Starch + PBAT	16.5 ± 2.4	311 ± 67	2.14 ± 0.25	8.90 ± 0.52	12.73 ± 1.98	5181 ± 1992

^1^ n = 5; average values and standard deviations are calculated. ^2^ n = 5; average values and standard deviations are calculated; orientation sample: penetration at outer side. ^3^ n = 10; average values and standard deviations are calculated.

**Table 3 polymers-14-02569-t003:** Results of gas barrier characterization (orientation samples: transmission rates are measured from outside to inside, inside = seal side).

Samples	OTR 0% RH, 23 °C (cc m^−^^2^ d^−1^) (n = 1)	OTR 50% RH, 23 °C (cc m^−^^2^ d^−1^) (n = 1)	WVTR 100% RH, 38 °C (g m^−^^2^ d^−1^) (n = 1)
1. Coated paper 1	3564	NA	29.1
2. Coated paper 2	2718	NA	>1000
3. Cellulose 1	0.40	3.65	187
4. Cellulose 2	0.34	5.78	58.8
5. Pilot extrusion PHBV	50.6	NA	36.8
6. Pilot extrusion PBS	122	NA	67.9
7. PBS	306	NA	420
8. PLA	519	NA	274
9. PLA + PBAT	2725	NA	1095
10. Starch + PBAT	1472	NA	624

**Table 4 polymers-14-02569-t004:** Results of seal characterization.

Samples	T_initiation_ ^1^ (°C)	T_max strength/2_ ^1^ (°C)	Seal Strength_max_ ^1^ (N mm^−1^)	T_initiation_ ^2^ (°C)	T_max. strength_ ^2^ (°C)	T_window_ ^2^ (°C)	Hot Tack Strength_max_ ^2^ (N mm^−1^)
1. Coated paper 1	100	105	0.40 ± 0.05	105	140	0	0.08 ± 0.00
2. Coated paper 2	80	85	0.49 ± 0.03	70	100	110	0.41 ± 0.02
3. Cellulose 1	115	115	0.11 ± 0.01	95	145	35	0.13 ± 0.01
4. Cellulose 2	75	85	2.69 ± 0.80	65	75	115	0.71 ± 0.02
5. Pilot extrusion PHBV	185	195	1.08 ± 0.09	115	135	40	0.37 ± 0.08
6. Pilot extrusion PBS	185	195	4.43 ± 1.50	125	150	0	0.12 ± 0.02
7. PBS	80	80	1.49 ± 0.06	65	70	20	0.40 ± 0.01
8. PLA	85	95	1.15 ± 0.05	75	140	70	0.33 ± 0.11
9. PLA + PBAT	85	95	0.29 ± 0.02	75	90	5	0.11 ± 0.01
10. Starch + PBAT	85	90	0.29 ± 0.01	75	80	5	0.13 ± 0.01

^1^ ASTM F88 (n = 3, average seal strength values and standard deviations are calculated). ^2^ ASTM F1921 (n = 3 average hot tack strength values and standard deviations are calculated).

**Table 5 polymers-14-02569-t005:** Maximized seal strengths of clean and contaminated seals.

Samples	Contamination	Predicted Value (N mm^−1^)	95% Confidence Interval (N mm^−1^)	Optimal Parameters (Seal Temperature, Time and Pressure)
Coated paper 1	Clean	0.40	0.24–0.38	135 °C, 1.5 s and 4 N mm^−2^
	Coffee powder	0.31	0.22–0.40	
	Sand	0.36	0.28–0.36	
	Water	0.40	0.19–0.45	
Coated paper 2	Clean	0.44	0.31–0.49	113 °C, 1.5 s and 4 N mm^−2^
	Coffee powder	0.37	0.25–0.44	
	Sand	0.46	0.32–0.50	
	Water	0.50	0.31–0.48	
Cellulose 1	Clean	0.18	0.15–0.19	180 °C, 0.4 s and 8 N mm^−2^
	Grated cheese	0.15	0.12–0.18	
Cellulose 2	Clean	3.40	2.90–3.60	180 °C, 0.4 s and 8 N mm^−2^
	Grated cheese	2.70	3.10–3.50	

**Table 6 polymers-14-02569-t006:** Average opacities Y (in %) and standard deviations (n = 4).

Samples	Y ± SD
1. Coated paper 1	81.9 ± 6.3
2. Coated paper 2	86.0 ± 2.7
3. Cellulose 1	11.5 ± 2.7
4. Cellulose 2	20.6 ± 0.3
5. Pilot extrusion PHBV	46.1 ± 0.9
6. Pilot extrusion PBS	24.8 ± 1.7
7. PBS	14.0 ± 0.3
8. PLA	7.9 ± 0.3
9. PLA + PBAT	98.7 ± 4.6
10. Starch + PBAT	16.1 ± 1.2

**Table 7 polymers-14-02569-t007:** Average water contact angles (WCA) (in °) and standard deviations (n = 15).

Samples	WCA ± SD
1. Coated paper 1	92.7 ± 4.0
2. Coated paper 2	85.1 ± 5.0
3. Cellulose 1	86.9 ± 3.4
4. Cellulose 2	89.6 ± 4.3
5. Pilot extrusion PHBV	95.2 ± 3.5
6. Pilot extrusion PBS	104.6 ± 4.3
7. PBS	84.2 ± 2.8
8. PLA	80.0 ± 4.3
9. PLA + PBAT	102.2 ± 4.3
10. Starch + PBAT	105.0 ± 1.6

## Data Availability

The data presented in this study are available on request from the corresponding author.

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
