# Peer review of "Characterizing Mechanical, Heat Seal, and Gas Barrier Performance of Biodegradable Films to Determine Food Packaging Applications"

_polymers, 2022, doi:10.3390/polym14132569_

Round 1

Reviewer 1 Report

Comment review

This paper reports the development of biodegradable packaging and its potential for food packaging. As a biodegradable material, cellulose has been widely used in the field of biodegradable packaging. Therefore, the development of cellulose-based degradable packaging has a wide application prospect. Also, the mechanical properties, gas barrier and sealing properties of the samples were characterized and samples of different coatings were made. However, this article still needs something worth discussing. My detailed comments are as follows:

1.      In introduction section, more background on the packaging materials should be provided with supporting articles: Nanomaterials 10 (1), 150, 2020; Development and characterization of food packaging bioplastic film from cocoa pod husk cellulose incorporated with sugarcane bagasse fibre; etc.

2.      For different series of samples, it is better to provide some digital photos of the samples and the samples for packaging applicaions, which is better for the understanding of readers.

3.      In this paper, the relationship between air permeability, moisture permeability and coating thickness should be illustrated, in that way the influence of coating on barrier property can be fully explained.

4.      Hydrophobic properties has a great influence on food packaging, so hydrophilic and hydrophobic properties of materials should be further characterized by relevant tests. Supplementary testing of the water contact Angle of the material is recommended. Please refer and cite: Superhydrophobic modification of cellulose and cotton textiles: Methodologies and applications

5.      There are so many different types of cellulose like CNC, CNW or CNF. Due to the great difference in properties between different types and sources of cellulose, the type, origin or source of cellulose should be described in detail in the article. In addition, more introductin on this point should be provided with supporting of necessary articles: A review on raw materials, commercial production and properties of lyocell fiber

6.      The light transmittance of materials should be supplemented to fully show the light transmittance of materials and reflect the appearance and decoration performance of packaging.

7.      Since there is no relevant data in this paper to explain the degradation performance of materials, data should be added to explain the degradation performance of materials.

8.      Typical stress-strain curves should be provided.

9.      Some figures should be modified to have better resolution and readability.

10.   Conclusions section is too complecated. Authors are suggested to rewrite this section in a more concise and logic way.

11.   There are still some formatting errors and syntax errors that need to be carefully corrected.

Reviewer 2 Report

L56, 57, 82, 83 Remove =

Please also add number of replication in Method section

L129 Avoid starting the sentence with number

L149 remove to

Add statistical analysis in method section

Table 2 Add statistical analysis and labels. The table heading should include Tensile, Puncture, Tear resistance for easier to understand the data

L286 Add more discussion e.g., Previous research investigated properties of PBAT blend polymers which were majorly influenced by compatibility between polymers and morphology including microstructures and crystallinity (doi.org/10.1016/j.foodcont.2021.108541).

L344 Add more discussion e.g., Katekhong (2022) fabricated biodegradable polymers with superior oxygen barrier, having oxygen permeability values less than 5 cm3.mm/m2.day.atm (doi.org/10.1016/j.foodchem.2021.131709).

L345-346 It is too short for one paragraph.

L349 Add more discussion e.g., High permeation of films is required for moisture and gas transfer in agricultural products to avoid accumulation of saturated moisture which leads to fungal growth and avoid anoxic condition (doi.org/10.1111/ijfs.15816).

Round 2

Reviewer 1 Report

Authors have addressed all the issues well. An acceptance is suggested.

Reviewer 2 Report

The manuscript has been improved.